# Back to normal? The health care situation of home care receivers across Europe during the COVID-19 pandemic and its implications on health

Michael Bergmann[1,2☺]*, Melanie Wagner[2☺]

1 Munich Research Institute for the Economics of Aging and SHARE Analyses (MEA-SHARE), Munich, Germany, 2 SHARE Berlin Institute (SBI), Berlin, Germany

☺ These authors contributed equally to this work.
* mbergmann@share-berlin.eu

**Data Availability Statement:** This paper uses data from the Survey of Health, Ageing and Retirement in Europe (SHARE). The data used in this study are available to all researchers for purely scientific

## Abstract

The COVID-19 pandemic began impacting Europe in early 2020, posing significant challenges for individuals requiring care. This group is particularly susceptible to severe COVID-19 infections and depends on regular health care services. In this article, we examine the situation of European care recipients aged 50 years and older 18 months after the pandemic outbreak and compare it to the initial phase of the pandemic. In the descriptive section, we illustrate the development of (unmet) care needs and access to health care throughout the pandemic. Additionally, we explore regional variations in health care receipt across Europe. In the analytical section, we shed light on the mid- and long-term health consequences of COVID-19-related restrictions on accessing health care services by making comparisons between care recipients and individuals without care needs. We conducted an analysis using data from the representative Corona Surveys of the Survey of Health, Ageing and Retirement in Europe (SHARE). Our study examines changes in approximately 3,400 care-dependent older Europeans (aged 50+) interviewed in 2020 and 2021, comparing them with more than 45,000 respondents not receiving care. The dataset provides a cross-national perspective on care recipients across 27 European countries and Israel. Our findings reveal that in 2021, compared to the previous year, difficulties in obtaining personal care from someone outside the household were significantly reduced in Western and Southern European countries. Access to health care services improved over the course of the pandemic, particularly with respect to medical treatments and appointments that had been canceled by health care institutions. However, even 18 months after the COVID-19 outbreak, a considerable number of treatments had been postponed either by respondents themselves or by health care institutions. These delayed medical treatments had adverse effects on the physical and mental health of both care receivers and individuals who did not rely on care.

purposes upon request on the SHARE website (https://share-eric.eu/data/data-access). The respective DOIs are given in the manuscript.

**Funding:** The authors received no specific funding for this work.

**Competing interests:** The authors have declared that no competing interests exist.

## Introduction

The COVID-19 pandemic started to hit European countries at the beginning of 2020. In the first phase of the pandemic, this has become problematic in particular for those in need of care. They often have chronic health conditions that make them predisposed to severe COVID-19 infections and to regular (health)care. While the dramatic situation of nursing home residents in the first phase of the pandemic is still present to most of us, related epidemiological control measures (physical distancing, stay-at-home requirements etc.) temporarily installed in almost all European countries have also greatly impacted those in need of care who live and are cared for at their own homes (the non-institutionalized). In fact, the vast majority of care receivers in Europe are living at home where care is mainly provided by relatives and friends [1] and to a smaller extent by formal care providers [2].

Appropriate access to health care as well as long-term care, either formal or informal, is a crucial prerequisite to preserve or improve health [3]. If access to health care is inadequate, unmet needs might have adverse consequences, such as poorer health or increasing (health) inequalities [4–7]. With the outbreak of the COVID-19 pandemic in early 2020, health care facilities in many European countries became overwhelmed by the crisis and thus had great difficulties in satisfying the needs of care receivers [8–10]. Existing studies from the first phase of the pandemic further showed that care receivers often discontinued paid home services because of fear of infection or because they were advised to do so [11–14]. Also, migrant care workers returned to their home countries and later could not return because of closed borders [15, 16]. In addition, day-care facilities where care receivers can spend some of their time often had to close [11] and health services and treatments experienced disruptions due to installed epidemiological control measures [17–21]. There is also evidence that help and support by closer family and friends increased during this phase [12, 22–25] as those had to compensate for the reduced supply of formal care providers. As health care services were often unavailable, care receivers forwent medical treatments [4, 26–29] and they also showed indications of physical and mental health problems [22, 30, 31].

Despite this information about care receivers at the beginning of the pandemic, little is known about the mid- and long-term implications of the COVID-19 pandemic regarding care and health care receipt of older Europeans over the course of the crisis. This is where our article wants to contribute. We build on previous research on (restricted) access to health care services for people aged 50+ in Europe [27, 32] that mainly focus on the determinants of unmet health care needs. Based on this research, the main objectives of this article are (1) to examine changes in terms of care receipt as well as access to health care during the pandemic and (2) to analyze the consequences of persistent unmet needs on physical and mental health for care receivers vis-à-vis people not relying on care. More precisely, we want to answer the following research questions, first referring to the situation of care recipients in Europe in summer 2021, about 18 months after the outbreak of the pandemic:

1. How was the situation of home care receivers in 2021 compared to the first phase of the pandemic with regard to the prevalence of care receipt (in general and by type of caregiver) and the perceived difficulties in receiving care?

2. How did the health care situation of older Europeans change during the pandemic and to what extent have medical treatments been deferred?
   Moreover, we investigated health-related outcomes for care recipients and respondents who did not receive home care. In this respect, we want to answer the following research questions:

3. Did the access to and use of health care services differ between receivers of care and persons not relying on care in 2020 and 2021?

4. How did delayed or postponed treatments affect older people's physical and mental health and were care recipients and people not relying on care affected differently?

## Data and methods

In the following analyses, we used released data from the first and second SHARE Corona Survey [SCS1 and SCS2; 33, 34] that were fielded, respectively, during summer 2020 and 2021 in 28 countries (Austria, Belgium, Bulgaria, Croatia, Cyprus, Czech Republic, Denmark, Estonia, Finland, France, Germany, Greece, Hungary, Israel, Italy, Latvia, Lithuania, Luxembourg, Malta, Netherlands, Poland, Portugal, Romania, Slovakia, Slovenia, Spain, Sweden and Switzerland). The SHARE Corona Survey is a Computer Assisted Telephone Interview (CATI), which was created in reaction to the COVID-19 crisis, collecting data on the living situation of people who are 50 years and older during the pandemic across Europe and Israel [see 35 for details on the background, sampling adaptions and fieldwork design aspects in the SCS1]. While the gross sample of the first SHARE Corona Survey consisted of all respondents that were eligible for the regular SHARE Wave 8 and did not become ineligible until start of fieldwork in June 2020, the second SHARE Corona Survey re-interviewed respondents of the first SHARE Corona Survey to enable the examination of changes between the start of the pandemic and the situation about one and a half years later. Oral consent, emphasizing the voluntary nature of participation and the confidentiality of data (e.g. researchers have no access to information that could identify individual participants during or after data collection), has been obtained from all participants. Respondent information from the regular SHARE panel study [36] was added during our analyses to provide long-term information on stable respondent characteristics [37–44]. The average response rate based on eligible respondents participating in the first SHARE Corona Survey was 79 percent. In the second SHARE Corona Survey, an average retention rate (excl. recovery of respondents) of 86 percent was achieved. To avoid selectivity, our analyses are based on 48,058 respondents aged 50 years and older who participated in both SHARE Corona Surveys.

In our analyses, we focused on informal care receipt. This was defined by the following question in the first SHARE Corona Survey: "Did you regularly receive home care before the outbreak of Corona?" In the second SHARE Corona Survey, the question wording was slightly adapted to capture the same period and to avoid country differences in the development of the pandemic: "During the last three months, did you regularly receive home care, provided by someone not living in your household?" This question was followed in the second SHARE Corona Survey by a request to indicate the provider of care receipt (if applicable). The list of care providers included one's own children, one's own parents, other relatives, other nonrelatives like neighbors, friends or colleagues, and professional care providers. This offers additional information that was not available in the first SHARE Corona Survey. Moreover, we used the respondents' answers on possible difficulties receiving personal care for our descriptive analyses. Also, here the time reference was adapted in the second compared to the first SHARE Corona Survey: "Since the outbreak of Corona (SCS1)/During the last three months (SCS2), did you face [more; only SCS1] difficulties in getting the amount of home care that you need?"

For analyzing problems regarding the continuation of medical treatments since the outbreak of the pandemic (SCS1)/compared to three months ago (SCS2), we further included

three dichotomized variables: whether a medical treatment was canceled by the respondents themselves because of being afraid of getting infected; whether a planned appointment for a medical treatment was postponed by the doctor or medical facility; and whether an appointment for a medical treatment was denied. To analyze the cumulated effect of delayed medical treatments on health, we used a dichotomous variable that is 0 when no medical treatments were deferred in the first and second SHARE Corona Survey and 1 when a delayed medical treatment was reported either in the first or second SHARE Corona Survey. With this operationalization, we took care of the possibility that medical treatments that had been deferred in the beginning of the pandemic still could affect health in the medium or long run.

To explore the direct and indirect effects of the pandemic, we included several indicators that measure changes in respondents' physical and mental health during the COVID-19 pandemic. For physical health, we used the following question from the second SHARE Corona Survey: "If you compare your health now to three months ago, would you say your health has improved, stayed about the same, or worsened?". To measure mental health, we used an additive index based on indications of anxiety ("In the last month, have you felt nervous, anxious, or on edge?"), depression ("In the last month, have you been sad or depressed?") and sleeping problems ("Have you had trouble sleeping recently?"). We then generated dichotomized variables that indicate a worsening of respondents' self-rated physical and mental health in case respondents confirmed that their health strains have increased since the outbreak of the pandemic: "Was that less so, about the same, or more so than during the first wave?".

Covariates that could confound the relationship with care receipt and health changes were selected according to previous studies in this area [27, 45, 46] and included sociodemographic characteristics as well as living conditions. We used the respondent's sex (0: male, 1: female) and the age at interview. Further, we coded the level of education attained based on the Internal Standard Classification of Education 1997 (ISCED-97). Respondents were then grouped into three categories: primary education (ISCED-97 score: 0–2), secondary education (ISCED-97 score: 3), and post-secondary education (ISCED-97 score: 4–6). Further, we used information on the respondents' type of living area (0: rural area, 1: urban area like a large town or big city), household composition (0: living with a partner, 1: living alone), employment status (0: not employed, 1: employed, incl. self-employment), and their economic status by a question that asked the degree to which respondents can make ends meet (0: with great/some difficulty, 1: fairly easily/easily). Finally, we included country dummies to account for regional differences, e.g., with respect to the national health care system. Table 1 provides an overview of the variables used in the analyses including summary statistics for both surveys (if applicable).

To address our research questions, we first descriptively explored changes in the prevalence of receiving home care as well as the access to and use of health care services across European regions, taking into account institutional and cultural similarities as well as diverse developments of the COVID-19 crisis [47–49], in summer 2021 compared to the first phase of the pandemic. Afterwards, we analyzed differences in unmet health care needs between care receivers and persons not receiving care as well as consequences of such unmet needs on physical and mental health. To do so, we used average adjusted predictions and marginal effects based on multivariate logistic regression models, which control for the respondent characteristics mentioned above [50, 51]. Generally, we estimate a logistic model:

$$\log\left(\frac{p_i}{1 - p_i}\right) = \beta_0 + \beta_1 x_{1i} + \cdots + \beta_p x_{pi},$$

**Table 1. Descriptive statistics.**

| Variables | SCS1 (2020) | N | SCS2 (2021) | N |
|---|---|---|---|---|
| *Dependent variables* | | | | |
| Care receipt | 4.9 | 48,028 | 7.2 | 48,005 |
| Perceived difficulties in receiving home care | 21.5 | 2,598 | 4.7 | 3,424 |
| Home care received from children | - | - | 41.6 | 3,073 |
| Home care received from parents & relatives | - | - | 11.9 | 3,259 |
| Home care received from non-relatives | - | - | 11.7 | 3,414 |
| Home care received from professional care providers | - | - | 69.3 | 3,493 |
| Forwent medical treatment | 11.7 | 48,017 | 9.0 | 47,961 |
| Postponed medical appointment | 25.8 | 48,013 | 12.0 | 47,923 |
| Denied medical appointment | 5.5 | 48,017 | 5.4 | 47,954 |
| Worsened physical health | - | - | 13.3 | 48,009 |
| Worsened mental health | - | - | 20.4 | 48,017 |
| *Control variables* | | | | |
| Age at interview | 67.1 (9.7) | 48,058 | 67.2 (10.0) | 48,058 |
| Gender: female | 54.1 | 48,058 | 54.3 | 48,058 |
| Educational level: primary | 36.3 | 47,333 | 34.1 | 47,333 |
| Educational level: secondary | 38.3 | 47,333 | 38.6 | 47,333 |
| Educational level: post-secondary | 25.5 | 47,333 | 27.3 | 47,333 |
| Household composition: living alone | 33.7 | 48,058 | 32.7 | 48,058 |
| Type of living area: urban | 35.4 | 46,326 | 35.6 | 46,414 |
| Make ends meet: (fairly) easily | 68.4 | 46,673 | 71.1 | 46,839 |
| Work status: employed (incl. self-employed) | 33.3 | 48,008 | 34.5 | 47,994 |
| *Country groups* | | | | |
| Northern European countries | 4.1 | | 4.5 | |
| Western European countries | 42.5 | | 43.7 | |
| Southern European countries | 32.8 | 48,058 | 32.2 | 48,058 |
| Eastern European countries | 19.1 | | 18.3 | |
| Baltic States | 1.5 | | 1.3 | |

Note: Entries are in percent (range: 0–100), except age at interview that is in years (range: 51–105 years).

Northern European countries: *Denmark, Finland, Sweden*; Western European countries: *Austria, Belgium, France, Germany, Luxembourg, Netherlands, Switzerland*; Southern European countries: *Croatia, Cyprus, Greece, Israel, Italy, Malta, Portugal, Slovenia, Spain*; Eastern European countries: *Bulgaria, Czech Republic, Hungary, Poland, Romania, Slovakia*; Baltic States: *Estonia, Latvia, Lithuania*.

Source: SHARE Wave 8 COVID-19 Survey 1, release 8.0.0 and SHARE Wave 9 COVID-19 Survey 2, release 8.0.0 (balanced sample of 48,058 respondents who participated in both SCS1 and SCS2; weighted).

which can be rewritten in the probability scale as

$$\Pr(y_i = 1 \mid x) = \frac{1}{1 + e^{-\left(\beta_0 + \beta_1 x_{1i} + \cdots + \beta_p x_{pi}\right)}}.$$

The marginal effect for a continuous covariate $x_1$ is then given by the expression:

$$\frac{\partial \Pr(y_i = 1 \mid x)}{\partial x_1} = \beta_1 \frac{e^{\beta_0 + \beta_1 x_{1i} + \cdots + \beta_p x_{pi}}}{\left(1 + e^{-(\beta_0 + \beta_1 x_{1i} + \cdots + \beta_p x_{pi})^2}\right)}.$$

The direction of the change is given by the sign of $\beta_1$. Further, it is evident that the effect of $x_1$ depends on the value of all other covariates in the model. With a dichotomous independent

variable, the marginal effect is simply the difference in the adjusted predictions (in the probability scale) for the two groups, for example, between respondents who delayed a medical treatment and those who did not. Average marginal effects compute the average of the marginal effects for each case, first treating all respondents as though they delayed a medical treatment and then as though they did not, leaving all other independent variable values as is.

With this approach, we are able to compare care receivers and respondents not receiving care that have identical values on all covariates included in the model, allowing to attribute delayed medical treatments and care receipt with much more confidence as the respective cause of differences in the probabilities of reporting physical and mental health strains or not. All analyses were run on a balanced sample of 48,058 respondents who participated in both the first and the second SHARE Corona Survey to allow proper comparisons that do not suffer from selective attrition. Small differences in numbers between the two surveys are due to slightly different item nonresponse rates of participating respondents. We used Stata 17.0 with calibration weights provided by the SHARE coordination team.

## Results

### The situation of care recipients during the pandemic

The overall prevalence of receiving home care across Europe rose between the first phase of the pandemic in 2020 and summer 2021 (see Fig 1). While the 2020 survey found that on average, 4.9 percent (n = 2,611) of all respondents older than 50 years received home care from others outside their own household, the prevalence increased to 7.2 percent (n = 3,440) in the 2021 survey. Regional variation showed the strongest relative increase in Western European countries (Austria, Belgium, France, Germany, Luxembourg, Netherlands, Switzerland), the Baltic States (Estonia, Latvia, Lithuania), and in Southern Europe (Croatia, Cyprus, Greece, Israel, Italy, Malta, Portugal, Slovenia, Spain). However, the increase was also considerable in Eastern Europe (Bulgaria, Czech Republic, Hungary, Poland, Romania, Slovakia). In contrast, no increase was found in Northern Europe (Denmark, Finland, Sweden).

When further investigating the age and health distribution of care receivers (see S1 Table) it can be seen that the absolute level of receiving home care was much higher for older respondents over 80 years (19.1% in SCS1 and 25.7% in SCS2) or suffering from limitations in activities of daily living (ADL; 22.8% in SCS1 and 25.7% in SCS2) compared to younger respondents between 50 and 65 years (1.3% in SCS1 and 2.5% in SCS2) or without ADL limitations (3.0% in SCS1 and 5.3% in SCS2). In addition, the relative increase of home care received by respondents between 2020 and 2021 was much stronger for younger people aged 50–64 (+92%) as well as those without limitations in ADL (+77%) than for older people over 80 years (+35%) or those suffering from ADL limitations (+13%). This indicates that especially younger, less limited people received again a higher amount of home care in 2021 than at the beginning of the COVID-19 pandemic when the focus was clearly on the most vulnerable groups of old and frail people.

Next, we explored the perceived difficulties in receiving home care during the different phases of the pandemic. Fig 2 shows the share of care recipients who reported that they faced difficulties in receiving care at the beginning of the COVID-19 pandemic in 2020 and one year later in 2021 by geographical regions. While in 2020 about 21 percent of all care recipients reported difficulties in receiving care, this share dropped substantially to less than 5 percent in 2021. This strong decrease was most pronounced in Southern and Western Europe, likely due to the severe problems of the national health care system and the subsequent strict epidemiological control measures in these regions at the beginning of the pandemic [52]. To underpin this finding with numbers: While nearly every third care recipient in Southern European

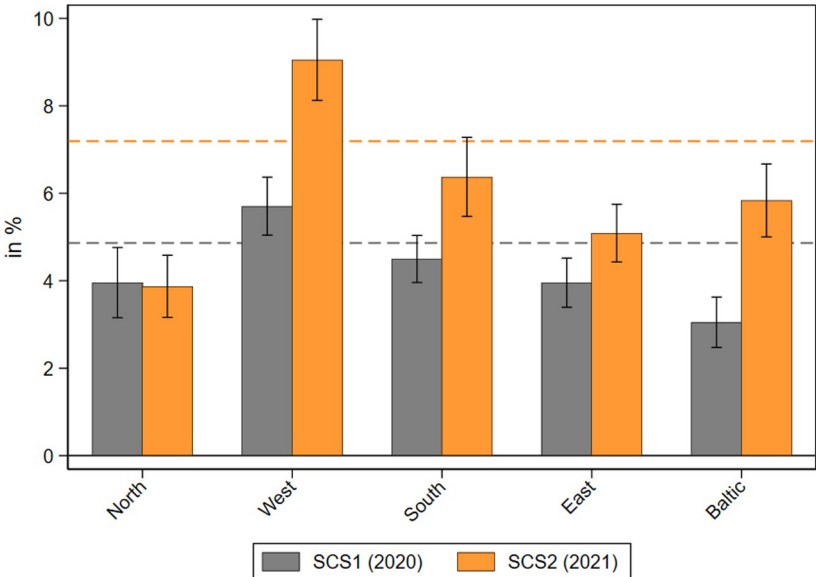

**Fig 1. Prevalence of receiving home care during the pandemic.** Data: SHARE Wave 8 COVID-19 Survey 1 and SHARE Wave 9 COVID-19 Survey 2, Release 8.0.0 (n = 48,028 and 48,005, respectively; weighted) with 95% confidence intervals.

countries reported difficulties in receiving care in the first phase of the pandemic, this was only done by every fifteenth care recipient (i.e. less than 7%) one year later. In Western Europe the percentage of care receivers who reported problems decreased even more from about 21 percent in 2020 to only slightly over 3 percent in 2021.

Home care in 2021 was mainly provided by professional care providers (69%) and by children (42%), while other relatives and nonrelatives (12% each) contributed to a lesser extent

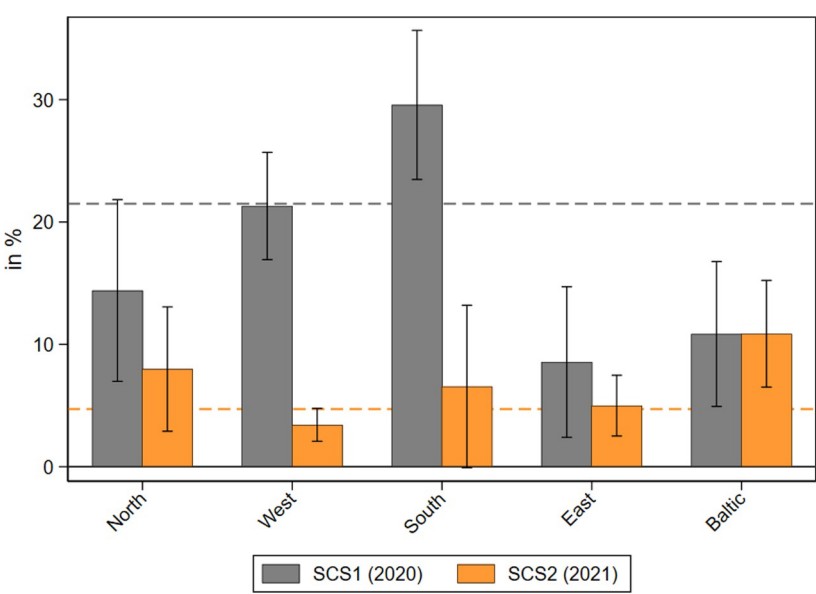

**Fig 2. Perceived difficulties in receiving home care during the pandemic.** Data: SHARE Wave 8 COVID-19 Survey 1 and SHARE Wave 9 COVID-19 Survey 2, Release 8.0.0 (n = 2,598 and 3,424 respectively; weighted) with 95% confidence intervals.

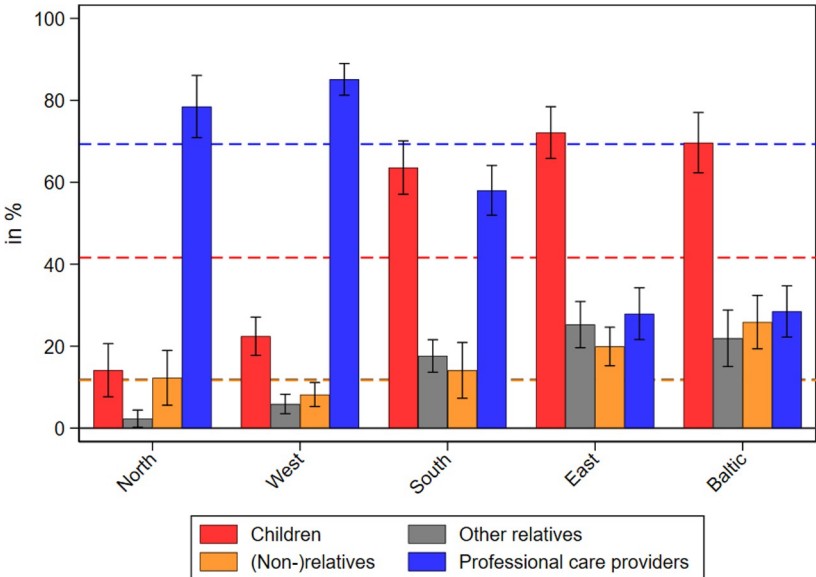

**Fig 3. Care receipt by different care providers in the second SHARE Corona Survey (2021).** Data: SHARE Wave 9 COVID-19 Survey 2, Release 8.0.0 (n = 3,073–3,493; weighted) with 95% confidence intervals. Dashed lines indicate means of care receipt across all countries.

(see Fig 3). Moreover, Fig 3 reveals large regional differences: Professional care providers accounted for the bulk of care provision in Western (85%) and Northern European countries (78%), while children played a much more prominent role in Eastern Europe (72%) and the Baltic States (70%). In Southern Europe, the picture was more mixed with children and professional care providers sharing responsibilities in rather similar parts (64% and 58%, respectively). As multiple answers were possible, the percentages do not sum up to 100. The most frequent pattern was care received from children and professional care providers. From all respondents receiving care in the second SHARE Corona Survey, about 15 percent received care from these two providers in parallel. All other patterns were found less frequently. Overall, more than 70 percent of respondents who stated to receive home care in the second SHARE Corona Survey only named one care provider, while about 20 percent named two. More than two providers were mentioned by only 5 percent of all care receivers.

## Unmet health care needs during the pandemic

The last section showed that home care receipt in general seemed less problematic in most European regions in 2021 compared to the first phase of the pandemic. In this paragraph, we check for possible long-term effects of the pandemic and its accompanying control measures regarding access to the national health care system. Therefore, we analyzed whether older respondents aged 50 years and over had problems in accessing medical treatments and if there were differences between those relying on personal care and those not. Fig 4 shows that in the second SHARE Corona Survey in 2021, about 9 percent of the respondents said that they forwent medical treatments (canceled by themselves) because of fear of infection. This is an overall reduction of about 3 percentage points compared to the beginning of the COVID-19 pandemic one year before. Differences were most pronounced in Southern and Eastern Europe and in the Baltic States, where the gaps reached a significant level.

Differences were even larger with regard to postponed medical appointments canceled by the health care institutions (see Fig 5). Here, the share of respondents who had a medical

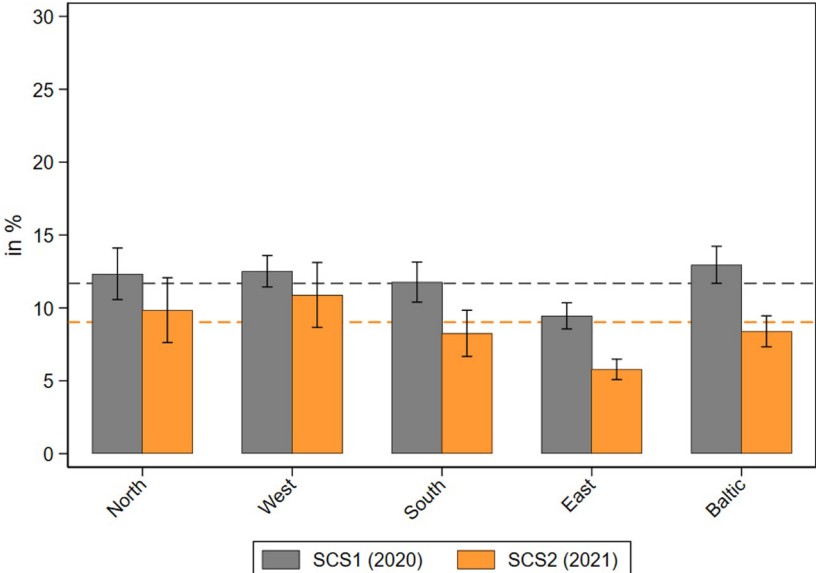

**Fig 4. Medical treatments forwent by respondents during the pandemic by geographical regions.** Data: SHARE Wave 8 COVID-19 Survey 1 and SHARE Wave 9 COVID-19 Survey 2, Release 8.0.0 (n = 48,017 and 47,961 respectively; weighted) with 95% confidence intervals.

appointment that the doctor or medical facility decided to postpone due to COVID-19 dropped from about 26 percent in 2020 to 12 percent one year later. This significant divergence was found across all regions, again pointing to an improved health care situation in 2021 compared to the first phase of the pandemic.

To assess the relevance of this decrease, we applied a robustness check by excluding forwent or postponed visits to specialists that also include dentists in the question text. Missed visits in this mutual category are likely to be mainly driven by postponed dentist check-ups [53], which might not have the same relevance for health outcomes in the longer run compared to a necessary operation or therapy. However, our findings revealed that although the absolute level of forwent and postponed medical treatments clearly decreased when excluding specialist visits (incl. dentists), unmet needs were still substantially lower across Europe in 2021 compared to one year before. Excluding this category would hence not change the interpretation of our findings regarding canceled medical treatments by the respondents themselves or by care facilities (see S1 and S2 Figs).

With respect to denied medical appointments canceled by health care institutions, the differences were much smaller, as was the absolute level (see Fig 6). About 5 percent of all respondents reported that they asked for an appointment for medical treatment but did not get one, both in 2020 and 2021. There was one exception: In the Baltic States, the share of denied appointments was much higher in 2020 (about 10%) and nearly halved one year later, likely due to the quickly and centralized implemented measures of infection control in Estonia, Lithuania, and Latvia shortly after the outbreak of the pandemic [54].

## Access to and use of health care services for care receivers and persons not receiving care

We further analyzed whether the more vulnerable group of care receivers experienced larger problems in getting access to medical treatments than respondents who did not rely on care. Table 2 shows average adjusted predictions based on multivariate logistic regression models

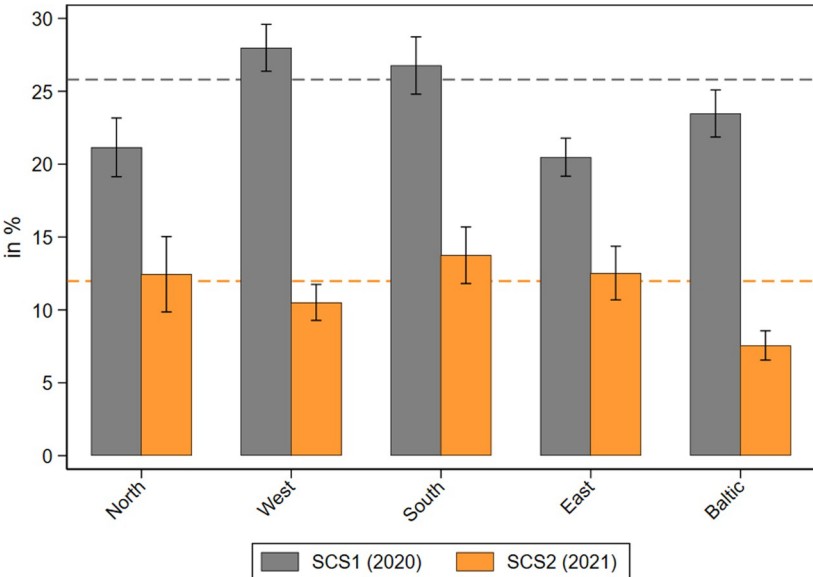

**Fig 5. Medical appointments postponed by health care institutions during the pandemic by geographical regions.**
Data: SHARE Wave 8 COVID-19 Survey 1 and SHARE Wave 9 COVID-19 Survey 2, Release 8.0.0 (n = 48,013 and 47,923 respectively; weighted) with 95% confidence intervals.

that control for the covariates described in the Data and Methods chapter. The depicted results reveal that in the first phase of the pandemic, care receivers compared to respondents not receiving care suffered significantly more often from postponed (31.1% vs. 25.7%; p = .006) and denied medical appointments (8.2% vs. 5.2%; p = .015) that have been canceled by the health care institution. In contrast, there was no significant difference in forwent medical treatments canceled by the respondents themselves (13.3% vs. 11.9%; p = .283) in 2020. This

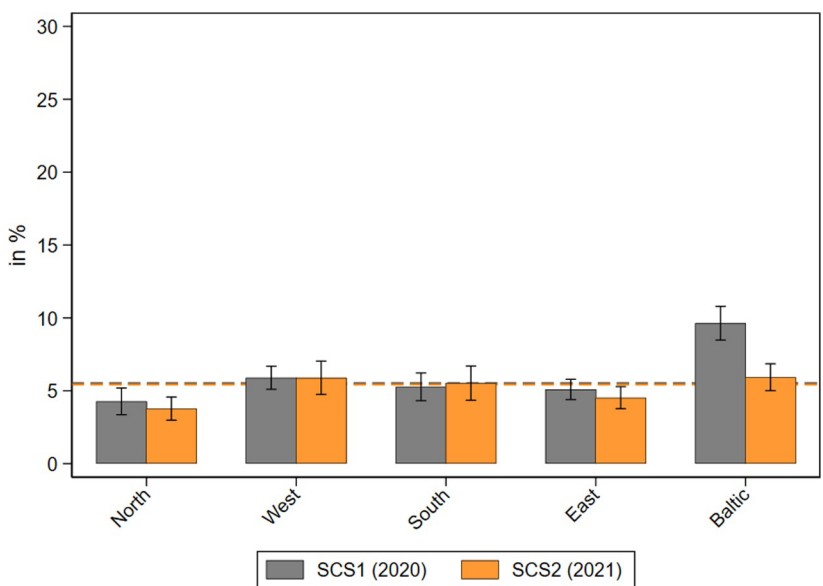

**Fig 6. Medical appointments denied by health care institutions during the pandemic by geographical regions.**
Data: SHARE Wave 8 COVID-19 Survey 1 and SHARE Wave 9 COVID-19 Survey 2, Release 8.0.0 (n = 48,017 and 47,954 respectively; weighted) with 95% confidence intervals.

**Table 2. Delayed medical treatments during the pandemic by care status of respondents.**

|  | Care recipient (yes/no) | SCS1 (2020) | | SCS2 (2021) | | Change over time (%-points) |
|---|---|---|---|---|---|---|
|  |  | Share (%) | Difference by care status (%-points) | Share (%) | Difference by care status (%-points) |  |
| Forwent medical treatment | Yes | 13.3 | 1.4 | 13.6 | 5.1** | 0.4 |
|  | No | 11.9 | | 8.6 | | -3.3*** |
| Postponed medical appointment | Yes | 31.1 | 5.3** | 15.8 | 4.0* | -15.3*** |
|  | No | 25.7 | | 11.8 | | -13.9*** |
| Denied medical appointment | Yes | 8.2 | 3.0* | 8.2 | 3.0 | 0.0 |
|  | No | 5.2 | | 5.2 | | 0.0 |

Significance: ***: p < .001

**: p < .01

*: p < .05 (significances based on average marginal effects (AMEs)).

Notes: Entries are adjusted predictions, controlled for sex, age, level of education, household composition, area of living, employment status, economic status and country of interview.

Source: SHARE Wave 8 COVID-19 Survey 1, release 8.0.0 and SHARE Wave 9 COVID-19 Survey 2, release 8.0.0 (n = 89,940, 89,909 and 89,939, respectively; weighted).

pattern changed in the second SHARE Corona Survey. In 2021, care receivers also reported forwent medical treatments due to fear of infection significantly more often than respondents not receiving care (13.6% vs. 8.6%; p = .009). When looking at the change over time (see last column) it gets clear that this difference was mainly caused by the decrease of forgone medical treatments in the group of respondents not receiving care, while there was even a small (insignificant) increase for care receivers. The general tendency of more frequent unmet needs for care receivers compared to respondents not receiving care can also be seen with regard to denied appointments for medical treatments. However, here the difference was somewhat smaller (3.0%) and did not change at all during the pandemic (see last column).

## The consequences of delayed medical treatments on physical and mental health

In a final step, we analyzed whether (and how) a delayed medical treatment during the pandemic–either because it was forgone by the respondent due to fearing an infection or postponed/denied by the care facility–affected physical and mental health in the second SHARE Corona Survey about 18 months after the outbreak of the pandemic. In addition, we investigated whether potential negative consequences were stronger for care receivers compared with respondents not receiving care. Like in the previous section, we used multivariate logistic regression models, now focusing on the interaction between delayed medical treatments and care status, while again controlling for a variety of relevant respondent background characteristics. In this respect, Table 3 depicts the results of the regression models with worsened physical and mental health as dichotomous dependent variables to analyze the implications of delayed treatments. As can be seen from the positive differences in the last column, both physical and mental health were perceived significantly worse by respondents when medical treatments or appointments were deferred during the pandemic. The largest differences were found with regard to medical appointments that were denied by care facilities. Further, mental health seemed to be affected somewhat stronger, which can also be noticed when comparing the absolute level of the share of respondents reporting worsened health. This share was between 7 and 10 percentage points higher than for physical health.

**Table 3. Marginal effects of delayed medical treatments during the pandemic on physical and mental health.**

| | Type of delayed health care service | | Share of worsened health (%) | Difference (%-points) |
|---|---|---|---|---|
| Worsened physical health | Forwent medical treatment | Yes | 15.5 | 2.7[**] |
| | | No | 12.8 | |
| | Postponed medical appointment | Yes | 15.0 | 2.6[**] |
| | | No | 12.5 | |
| | Denied medical appointment | Yes | 21.0 | 8.5[***] |
| | | No | 12.5 | |
| Worsened mental health | Forwent medical treatment | Yes | 26.1 | 6.8[***] |
| | | No | 19.4 | |
| | Postponed medical appointment | Yes | 24.0 | 4.9[***] |
| | | No | 19.1 | |
| | Denied medical appointment | Yes | 29.3 | 9.7[***] |
| | | No | 19.6 | |

Significance: ***: p < .001

**: p < .01

*: p < .05 (significances based on average marginal effects (AMEs)).

Notes: Entries are adjusted predictions, controlled for sex, age, level of education, household composition, area of living, employment status, economic status and country of interview.

Source: SHARE Wave 9 COVID-19 Survey 2, release 8.0.0 (n = 45,011–45,046; weighted).

In addition, Table 4 further specifies these findings by distinguishing between care receivers and respondents not receiving care. In this respect, it can be seen that, overall, there were no significant differences between these two groups when analyzing the relationship between delayed medical treatments during the pandemic and worsened health compared to three months before (see last column). Although the difference was positive in most cases, pointing to a larger marginal effect of delayed medical treatments for care receivers compared to respondents not receiving care (in particular concerning physical health), these differences did not reach a significant level. This contrasts with the observation that care receivers more frequently reported worsened health than respondents who did not receive care (see column "Share (%)"). That is, our findings here and in Table 2 point out that although respondents' care status shows a strong association with health deterioration (care receivers reported a worsening health consistently more often than respondents not receiving care), the negative consequences of delayed medical treatments 18 months after the outbreak of the pandemic were not significantly larger for care receivers.

## Discussion

This paper analyzed the (health) care situation of Europeans aged 50 years and older during the first one and a half years of the COVID-19 pandemic. By explicitly focusing on care recipients, we were able to analyze changes in the access to and the use of health care services to study the consequences of delayed medical treatments on their physical and mental health vis-à-vis respondents not relying on care. Based on the two waves of the SHARE Corona Survey in 2020 and 2021, we applied our analyses on a large representative sample of more than 45,000 respondents aged 50 years and older across 27 European countries and Israel.

Our findings revealed that the overall situation of care receivers improved in the sense that home care from outside the household was again more widely available in summer 2021 compared to one year before in nearly all European regions. There are two interpretations for this

**Table 4. Marginal effects of delayed medical treatments during the pandemic on physical and mental health by care status of respondents.**

| | Care receipt | Type of delayed health care service | | Share (%) | Difference (%-points) | Difference by care status (%-points) |
|---|---|---|---|---|---|---|
| **Worsened physical health** | yes | Forwent medical treatment | yes | 28.8 | 2.8 | 0.2 |
| | | | no | 26.0 | | |
| | no | | yes | 14.1 | 2.6[*] | |
| | | | no | 11.6 | | |
| | yes | Postponed medical appointment | yes | 30.3 | 5.8 | 3.9 |
| | | | no | 24.5 | | |
| | no | | yes | 13.3 | 1.9[*] | |
| | | | no | 11.4 | | |
| | yes | Denied medical appointment | yes | 38.1 | 13.2 | 5.7 |
| | | | no | 24.9 | | |
| | no | | yes | 18.9 | 7.6[***] | |
| | | | no | 11.3 | | |
| **Worsened mental health** | yes | Forwent medical treatment | yes | 35.5 | 4.1 | -2.8 |
| | | | no | 31.4 | | |
| | no | | yes | 25.2 | 6.9[***] | |
| | | | no | 18.4 | | |
| | yes | Postponed medical appointment | yes | 33.6 | 2.3 | -2.6 |
| | | | no | 31.3 | | |
| | no | | yes | 23.0 | 4.9[***] | |
| | | | no | 18.1 | | |
| | yes | Denied medical appointment | yes | 42.4 | 12.0 | 2.9 |
| | | | no | 30.5 | | |
| | no | | yes | 27.8 | 9.1[***] | |
| | | | no | 18.7 | | |

Significance: ***: $p < .001$

**: $p < .01$

*: $p < .05$ (significances based on average marginal effects (AMEs)).

Notes: Entries are adjusted predictions, controlled for sex, age, level of education, household composition, area of living, employment status, economic status and country of interview.

Source: SHARE Wave 9 COVID-19 Survey 2, release 8.0.0 (n = 45,002–45,037; weighted).

pattern: First, due to the vaccination campaign, which started at the end of 2020 and picked up speed in spring 2021 in most European countries, epidemiological control measures and restrictions that had been introduced could be relaxed so that social contacts as well as support in general (and care in particular) were possible again more easily [23]. This was particularly true in Western and Southern European countries, which had been hit hardest by the first wave of the pandemic in 2020 and hence had more restrictive control measures in place [52] compared to one year later when vaccination was available for large parts of the population. Moreover, there was presumably less fear of a severe SARS-CoV-2 infection for vaccinated individuals, especially for younger and less severely care-dependent people. This group again received a higher relative amount of home care in 2021 compared to the beginning of the pandemic when access to strictly necessary health care services was clearly restricted to the most vulnerable people.

A second methodological explanation is based on the question formulation that slightly changed between the first and second SHARE Corona questionnaire. Thus, while the latter survey in 2021 explicitly mentioned "professionals" in addition to relatives or friends

providing home care for those in need in the interviewer instructions, the former survey in 2020 did not. Thus, it might be the case that respondents in the second SHARE Corona Survey had a broader understanding of (formal and informal) home care, which might explain to some extent the higher percentage of care receipt in the second SHARE Corona Survey compared to the first. However, at the same time, it has to be considered that the second SHARE Corona Survey contains the explicit specification of "people from outside your home" from which home care was received. It is likely that this further clarification had a lowering effect compared to the first SHARE Corona Survey, where respondents had to interpret this based on previous questions. Overall, we thus believe that the found increase of home care receipt between 2020 and 2021 is rather substantive than due to a question wording effect. This view is backed up by our finding of significantly less frequently reported difficulties in receiving home care in 2021, when only about 5 percent of care receivers reported problems compared to the beginning of the pandemic, when more than 20 percent reported problems. In this respect, changes in question wording cannot easily justify the perceived improvement of the situation of care recipients, which again was most notable in Western and Southern European countries.

When looking at those who provided home care, our findings revealed large regional differences across Europe. While professionals were the main providers of care in Western and Northern European countries in 2021, children did the bulk of care work in Eastern Europe and the Baltic States. These findings are in line with theories and empirical findings that relate care use with family norms and welfare state arrangements [55–60]. While most countries in Northern and Western Europe are described as defamilialized (or individualistic), which is reflected in a high share of formal care services, Eastern Europe and the Baltic States are rather seen as family-oriented with a low share of formal care options [56, 57]. According to this literature, Southern European countries are usually classified as family-oriented and should therefore show lower levels of formal care use. However, our results indicate that children and professionals share the responsibility of care provision in Southern Europe. This might be explained by differences in the composition of countries that are summarized as "Southern Europeans". While earlier studies mainly focused on South-Western Europe, such as Spain, Italy, Portugal, and Greece [56, 57], our study also included countries from South-Eastern Europe (Croatia and Slovenia) as well as Israel. According to Saraceno [57], familialism can be further distinguished into different types, from which the type of "supported familialism" can go hand in hand with formal home care, while the other two ("familialism by default", "prescribed familialism") are primarily characterized by the absence of formal care options. In this respect, Israel, which shows a very high share of formal care services by professionals, and to some extent also Croatia might be classified by supported familialism, while in Italy and Greece, much higher shares of children providing care for their older parents were found.

Next, we investigated the health care situation in Europe with a particular focus on unmet health care needs. The descriptive findings showed that medical treatments were postponed less often by respondents fearing an infection as well as by health care institutions, both indicating a generally improved access to and use of necessary health care services across Europe in summer 2021 compared to one year before. While especially at the beginning of the pandemic health care institutions were overstained in some countries [9, 10] and/or implemented rather strict measures of infection control leading to cancelations or postponements of routine services [54], the situation generally improved during the pandemic when more and more people were vaccinated.

However, this general picture must be put into perspective when distinguishing between care receivers and persons not receiving care. Here, our analyses based on average adjusted predictions controlling for a broad range of relevant covariates showed that care recipients still

experienced significantly more problems in getting access to health care services one and a half years after the outbreak of the pandemic. This was true for postponed medical appointments canceled by health care facilities but also regarding forwent medical treatments canceled by respondents themselves. While differences between care recipients and people not receiving care persisted regarding postponed and denied medical treatments canceled by health care institutions, they became even larger with respect to medical treatments that have been forgone by respondents themselves because of fear of infection. This indicates that the situation at the beginning of the pandemic was similar (bad) for care receivers and respondents not receiving care with respect to the uncertainty linked to a SARS-CoV-2 infection. With progressing vaccinations, particularly respondents not relying on care reported less forwent medical treatments due to fearing an infection in 2021, while this was not the case for care receivers. That is, although the overall level of forwent medical treatments and postponed appointments decreased during the pandemic, care receivers still experienced significantly more problems in getting access to health care services, both because of fearing an infection but also because of the existing capacity problems of care facilities and doctors.

These delayed medical treatments had a significant negative impact on physical and mental health. In particular, medical appointments that have been denied by health care institutions have led to substantial worsened health. This result is in line with other studies showing a decline in health for certain population groups in the first phase of the pandemic [22, 31, 61] and increased morbidity [28]. However, the implications of delayed medical treatments on health outcomes did not significantly differ between respondents receiving versus not receiving care. That is, although care receivers were disadvantaged with regard to access and use of health care services, this did not yield significantly stronger deteriorations of physical and mental health compared to people not relying on care. While negative consequences of delayed medical treatments on physical health were indeed somewhat more pronounced for care receivers than for people without care needs, the difference did not reach a significant level when controlling for relevant background characteristics. Thus, although other studies reported higher unmet health care needs in the first phase of the pandemic for economically disadvantaged older people [4, 62] or people with comorbidities [29] and poor health [63], this did not directly translate into substantive health inequalities between those in need of care and those not–at least not in the medium run about one and a half years after the outbreak of the pandemic. Whether this will change eventually remains to be seen. The fact that care recipients keep foregoing medical treatments and that these treatments are not generally caught up [64] gives at least some cause for concern. If delayed necessary treatments will not be caught up in a reasonable period, this might still lead to health deterioration in the long run. Therefore, further research in this area is needed to better assess the implications of delayed medical treatments for care receivers (but also for people not relying on care) over the next few years.

While the data from the first SHARE Corona Survey were collected at the very beginning of the pandemic and the second SHARE Corona Survey was fielded one year later during the pandemic, it became possible to compare short- and mid-term effects using harmonized panel data from full probability samples in 28 countries. Being able to study many home care recipients (more than 3,400 in 2021) from a cross-national perspective is a key strength of this article. Thus, we were able to evaluate the health consequences of the pandemic across Europe for one of the most vulnerable groups, the home care receivers in Europe.

However, like other studies our analyses have some limitations, too. We already mentioned the differences in questionnaire wording between the first and second SHARE Corona Surveys. This limits the conclusiveness to some extent, although we have good arguments to believe that our interpretation of a substantive recovery in care receipt instead of a pure question wording effect is correct: Most importantly, this trend is also mirrored in the parallel decrease of

difficulties in receiving care. Especially Western and Southern European countries that have been hit hardest by the first wave of the pandemic in 2020 and in most cases also showed the highest vaccination rates afterwards were able to relax some of the strict epidemiological control measures, which very likely resulted in the found increase of care receipt in 2021.

Another point of critique is that the fieldwork of both the SHARE Corona Survey in 2020 and in 2021 was during summer. This must be recognized regarding the absolute level of receiving care or reporting unmet health care needs. People might have forgotten how they felt during winter and/or lockdown periods. In this respect, our findings might depict a too optimistic picture of the situation in 2020 and 2021. Further, care receivers obviously differ from people not relying on care and thus could be on a different health trajectory irrespective of unmet health care needs. More research is thus needed to verify our results. In a similar way, it must be noticed that a large fraction of the SHARE respondents, although aged 50 years and older, are still in rather good health. Therefore, our analyses might underrepresent the actual prevalence of care receipt in Europe and severe health conditions. Linked to this, the specific age group of our sample must be carefully considered when drawing generalized conclusions based on our results.

Finally, it became clear that our results indicate some limitations of the country grouping, particularly regarding the descriptive part of the analyses. This reflects the questioning of standard welfare state characterizations [65, 66] in recent decades due to a lack of consensus on the classification of welfare states in Europe [67, 68]. These inconsistencies were probably even further increased by the COVID-19 crisis, which hit the countries in a comparable but unforeseeable way and hence might have attenuated the link between welfare organizations and country-specific arrangements of national health care systems. Further comparative research is required also here to shed light on this and related questions.

Despite these limitations, our study contributes to the existing literature by providing a cross-national overview of how care receipt by older Europeans aged 50 years and older changed during the pandemic and whether unmet health care needs lead to deteriorations of physical and mental health. Our results show that despite a general improvement in the health care situation between 2020 and 2021, difficulties in accessing medical treatments persisted for one and a half years after the outbreak of the pandemic, at least to some extent. Such delayed treatments and visits had an overall negative effect on physical and mental health but discriminated care receivers not significantly stronger than persons not relying on care. Nonetheless, care receivers are a highly vulnerable group that was particularly affected by the COVID-19 pandemic. In this respect, policymakers and health care institutions should pay special attention to those in need of care and emphasize the importance of catching up postponed medical treatments. Otherwise, negative health outcomes are still possible in the long run, which is why future (longitudinal) research in this area is crucially needed in our view.

## Supporting information

**S1 Table. Prevalence of receiving home care during the pandemic (in percent) by age and health groups.** Data: SHARE Wave 8 COVID-19 Survey 1 and SHARE Wave 9 COVID-19 Survey 2, Release version: 8.0.0 (n = 48,058, respectively; weighted) with 95% confidence intervals in brackets.
(DOCX)

**S1 Fig. Medical treatments (without check-up at a specialist, incl. a dentist) forwent by respondents during the pandemic by geographical regions.** Data: SHARE Wave 8 COVID-19 Survey 1 and SHARE Wave 9 COVID-19 Survey 2, Release 8.0.0 (n = 48,016 and 47,958

respectively; weighted) with 95% confidence intervals.
(TIF)

**S2 Fig. Medical appointments (without check-up at a specialist, incl. a dentist) postponed by health care institutions during the pandemic by geographical regions.** Data: SHARE Wave 8 COVID-19 Survey 1 and SHARE Wave 9 COVID-19 Survey 2, Release 8.0.0 (n = 48,012 and 47,921 respectively; weighted) with 95% confidence intervals.
(TIF)

## Author Contributions

**Conceptualization:** Michael Bergmann, Melanie Wagner.

**Data curation:** Michael Bergmann, Melanie Wagner.

**Formal analysis:** Michael Bergmann, Melanie Wagner.

**Investigation:** Michael Bergmann, Melanie Wagner.

**Methodology:** Michael Bergmann, Melanie Wagner.

**Software:** Michael Bergmann, Melanie Wagner.

**Validation:** Michael Bergmann, Melanie Wagner.

**Visualization:** Michael Bergmann, Melanie Wagner.

**Writing – original draft:** Michael Bergmann, Melanie Wagner.

**Writing – review & editing:** Michael Bergmann, Melanie Wagner.

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
