## [Decision Letter · Decision Letter 0]

30 Aug 2023

PONE-D-23-16233Back to normal? The health care situation of home care receivers across Europe during the COVID-19 pandemic and its implications on healthPLOS ONE

Dear Dr. Bergmann,

Thank you for submitting your manuscript to PLOS ONE. After careful consideration, we feel that it has merit but does not fully meet PLOS ONE’s publication criteria as it currently stands. Therefore, we invite you to submit a revised version of the manuscript that addresses the points raised during the review process.

We look forward to receiving your revised manuscript.

Kind regards,

Mihajlo Jakovljevic, MD, PhD, MAE

Academic Editor

PLOS ONE

Journal Requirements:

Reviewers' comments:

Reviewer's Responses to Questions

**Comments to the Author**

1. Is the manuscript technically sound, and do the data support the conclusions?

Reviewer #1: Yes

Reviewer #2: Yes

2. Has the statistical analysis been performed appropriately and rigorously? 

Reviewer #1: Yes

Reviewer #2: Yes

3. Have the authors made all data underlying the findings in their manuscript fully available?

Reviewer #1: Yes

Reviewer #2: Yes

4. Is the manuscript presented in an intelligible fashion and written in standard English?

Reviewer #1: Yes

Reviewer #2: Yes

5. Review Comments to the Author

Reviewer #1: This is a cross-sectional study based on SHARE investigating changes in care receipt by Europeans aged 50+ during the COVID-19 pandemic and whether unmet healthcare needs lead to deteriorations of physical and mental health. It is a well-written study conducted by authors who knows a lot about the SHARE survey, which is reflected in the paper. However, it is a comprehensive study including four aims and could have benefitted from being shortened a bit down.

A couple of minor comments:

I prefer that strengths of the study are presented before limitations and not at the end of summary of results in the first part of the discussion.

Page 21 lines 515-517: Can you reformulate this part and perhaps insert the good arguments that you have of why your interpretation is correct.

Page 8 lines 211-214: should be reformulated to be easier to read.

Reviewer #2: The paper is well written and deserves publication in Plos One. More in detail, the paper focuses on an interesting topic, that is the effect of Covid-19 on health cares in Europe by examining changes in a large sample of care-dependent Europeans comparing them with individuals who did not receive care. In order to do this, the paper uses high quality data from SHARE survey led on the subset of Europeans aged 50 years or more and investigates changes in time using a series of marginal effects from logistic regressions.

The analysis is well performed and includes a complete set of explanatory variables, but few elements have to be addressed before final acceptance:

1) The presentation of the model has to be improved: the authors should summarize in a Table placed in the model section the list of dependent and explanatory variables. Indeed, in the discussion the reader experiences the risk of losing many details. In realizing that table the authors should clearly connect to the original wording of the questions used as dependent variables. In addition, descriptive statistics table are totally absent, because the paper starts with table 1 which directly introduces to the estimation of marginal effects.

2) The authors decompose the analysis in country groups. Literature provides suggestions in order to group countries in clusters on the basis of welfare states: see, for example, the classification of Esping-Andersen and similar efforts from other authors. My question is: have the authors taken into account that literature? And in case of positive response, is the classification used in the paper consistent with those findings? I would like to read more on this topic, because welfare organization is directly connected with health care systems.

6. PLOS authors have the option to publish the peer review history of their article (what does this mean?). If published, this will include your full peer review and any attached files.

Reviewer #1: No

Reviewer #2: **Yes: **Emiliano Sironi

---

## [Author Response · Author response to Decision Letter 0]

26 Sep 2023

PONE-D-23-16233

Back to normal? The health care situation of home care receivers across Europe during the COVID-19 pandemic and its implications on health

Dear editors and reviewers,

Thank you very much for your valuable comments and the opportunity to revise our manuscript. We carefully addressed your comments and believe that our paper has been considerably improved by this! Please find enclosed our detailed answers to the points you have raised ('Response to Reviewers'). All substantial changes towards the original manuscript are highlighted in the marked-up copy of the manuscript labelled 'Revised Manuscript with Track Changes'. The page and line numbers in our responses below refer to this version. In addition, we uploaded a cleaned version of the manuscript labelled 'Manuscript'.

Yours sincerely, 

Michael Bergmann (also on behalf of my co-author)

Response to Reviewers

Reviewer #1: 

This is a cross-sectional study based on SHARE investigating changes in care receipt by Europeans aged 50+ during the COVID-19 pandemic and whether unmet healthcare needs lead to deteriorations of physical and mental health. It is a well-written study conducted by authors who knows a lot about the SHARE survey, which is reflected in the paper. However, it is a comprehensive study including four aims and could have benefitted from being shortened a bit down.

Response: Thank you! We went carefully through the text and dropped two additional analyses: one on changes in care receipt by different care providers (lines 253-271) and another one on the type of forwent medical treatment (lines 321-341) to shorten the respective sections a bit. We hope that these cuts help the reader to follow the analyses of our four key questions, which we decided to keep. They are, in our view, needed to link the observed changes of care receipt and healthcare access during the pandemic (hypotheses 1 and 2) with the possible consequences on physical and mental health of care receivers and those not receiving care (hypotheses 3 and 4).

A couple of minor comments:

I prefer that strengths of the study are presented before limitations and not at the end of summary of results in the first part of the discussion.

Response: We shifted the respective sentence to the suggested position before the limitations and added a few more aspects which we consider as strengths of our article (lines 529-536). 

Page 21 lines 515-517: Can you reformulate this part and perhaps insert the good arguments that you have of why your interpretation is correct.

Response: Good point, maybe this was not clear enough. According to your suggestion, we added our arguments (i.e. parallel decrease of difficulties in receiving care, relaxed epidemiological control measures, comparable high vaccination rates), which very likely resulted in the recovery of care receipt in 2021 compared to the first wave of the pandemic in 2020 (lines 541-545).

Page 8 lines 211-214: should be reformulated to be easier to read.

Response: Thank you! We reformulated the sentence (lines 224-229).

Reviewer #2: 

The paper is well written and deserves publication in Plos One. More in detail, the paper focuses on an interesting topic, that is the effect of Covid-19 on health cares in Europe by examining changes in a large sample of care-dependent Europeans comparing them with individuals who did not receive care. In order to do this, the paper uses high quality data from SHARE survey led on the subset of Europeans aged 50 years or more and investigates changes in time using a series of marginal effects from logistic regressions.

The analysis is well performed and includes a complete set of explanatory variables, but few elements have to be addressed before final acceptance:

1) The presentation of the model has to be improved: the authors should summarize in a Table placed in the model section the list of dependent and explanatory variables. Indeed, in the discussion the reader experiences the risk of losing many details. In realizing that table the authors should clearly connect to the original wording of the questions used as dependent variables. In addition, descriptive statistics table are totally absent, because the paper starts with table 1 which directly introduces to the estimation of marginal effects.

Response: Thank you very much for this suggestion. We compiled a table (Table 1; see p. 6) including descriptive statistics for all dependent and control variables and using the same wording as in the other figures and tables. We hope that this makes it easier for the reader to follow both the analyses and the discussion.

2) The authors decompose the analysis in country groups. Literature provides suggestions in order to group countries in clusters on the basis of welfare states: see, for example, the classification of Esping-Andersen and similar efforts from other authors. My question is: have the authors taken into account that literature? And in case of positive response, is the classification used in the paper consistent with those findings? I would like to read more on this topic, because welfare organization is directly connected with health care systems.

Response: Thank you very much for raising this point! The rationale behind the applied grouping is based on several reasons, which we have added to the manuscript and that we will also explain in the following. First, although our analyses are based on a sample of care receivers that is larger than in most other studies, the observed number of respondents did not allow reasonable analyses on a country level (which would have been our preference). We tried to circumvent this problem by geographically grouping countries to measure the diverse developments of the pandemic across Europe on healthcare in general and care recipients in particular. As the pandemic, especially in the beginning, was characterized by local outbreaks and a clearly linked spreading, our aim was to reflect this in our country grouping, while also taking into account institutional and cultural similarities between European regions as well as different governmental responses to the COVID-19 crisis. We tried to make this clearer in the section on data and methods (lines 160-162). 

Second, standard welfare state characterizations (e.g. Esping-Andersen, 1990; Ferrera, 1996) have been criticized, inter alia, on the basis of substantial changes of welfare systems in recent decades and a related lack of consensus on the classification of welfare states in Europe (e.g. Buhr & Stoy, 2015; Powell et al., 2020). One explanation of this inconsistency highlights the incoherence across policy areas even within countries (e.g. Kasza, 2002). With respect to healthcare systems, recent findings hence could show the absence of clear clusters across European countries (e.g. Bertin et al., 2021). While we nevertheless agree with your notion that welfare organization is, in general, connected with health care systems, we further believe that the COVID-19 crisis hit the countries in a comparable but unforeseeable way, which might have further attenuated the link to country-specific arrangements of the healthcare system. Against this background, we already tried to link our findings with previous work that relate care use with family norms and welfare state arrangements (see our argumentation in lines 457-476), although it is clear that our country grouping has some limitations. We thus added a section to make this clearer and to highlight the need of more research in this direction (see lines 557-564).

Third, while the descriptive analyses on the change of care receipt and access to healthcare are based on a regional grouping of countries to better illustrate the development during the pandemic, our results of the consequences on physical and mental health were already based on regression models, which include country dummies to account for regional differences, e.g., with respect to the national health care system (see lines 146-147).

Literature:

Bertin, G., Carrino, L., Pantalone, M. (2021). Do standard classifications still represent European welfare typologies? Novel evidence from studies on health and social care. Social Science & Medicine 281 (114086). Doi: 10.1016/j.socscimed.2021.114086.

Buhr, D., Stoy, V. (2015). More than just welfare transfers? A review of the scope of Esping-Andersen's welfare regime typology. Social Policy and Society, 14 (2), pp. 271-285. Doi: 10.1017/S1474746414000542.

Esping-Andersen, G. (1990). The three worlds of welfare capitalism. Princeton: University Press.

Ferrera, M. (1996). The 'southern model' of welfare in social Europe. Journal of European Social Policy, 6 (1), pp. 17-37. 10.1177/095892879600600102.

Kasza, G.J. (2002). The illusion of welfare ‘regimes’. Journal of Social Policy, 31 (2), pp. 271-287. Doi: 10.1017/S0047279401006584. 

Powell, M.,Yörük, E., Bargu, A. (2020). Thirty years of the three worlds of welfare capitalism: a review of reviews. Social Policy & Administration, 54 (1), pp. 60-87. Doi: 10.1111/spol.12510.

---

## [Decision Letter · Decision Letter 1]

11 Oct 2023

Back to normal? The health care situation of home care receivers across Europe during the COVID-19 pandemic and its implications on health

PONE-D-23-16233R1

Dear Dr. Bergmann,

We’re pleased to inform you that your manuscript has been judged scientifically suitable for publication and will be formally accepted for publication once it meets all outstanding technical requirements.

Kind regards,

Mihajlo Jakovljevic, MD, PhD, MAE

Academic Editor

PLOS ONE

Additional Editor Comments (optional):

Reviewers' comments:

Reviewer's Responses to Questions

**Comments to the Author**

1. If the authors have adequately addressed your comments raised in a previous round of review and you feel that this manuscript is now acceptable for publication, you may indicate that here to bypass the “Comments to the Author” section, enter your conflict of interest statement in the “Confidential to Editor” section, and submit your "Accept" recommendation.

Reviewer #1: All comments have been addressed

Reviewer #2: All comments have been addressed

2. Is the manuscript technically sound, and do the data support the conclusions?

Reviewer #1: Yes

Reviewer #2: Yes

3. Has the statistical analysis been performed appropriately and rigorously? 

Reviewer #1: Yes

Reviewer #2: Yes

4. Have the authors made all data underlying the findings in their manuscript fully available?

Reviewer #1: Yes

Reviewer #2: Yes

5. Is the manuscript presented in an intelligible fashion and written in standard English?

Reviewer #1: Yes

Reviewer #2: Yes

6. Review Comments to the Author

Reviewer #1: The authors have answered all my comments and concerns from the first review. I suggest publication of the manuscript.

Reviewer #2: The paper now is ok. Authors have addressed my points.

7. PLOS authors have the option to publish the peer review history of their article (what does this mean?). If published, this will include your full peer review and any attached files.

Reviewer #1: No

Reviewer #2: **Yes: **Emiliano Sironi

---

## [Editor Report · Acceptance letter]

13 Oct 2023

PONE-D-23-16233R1 

Back to normal?
The health care situation of home care receivers across Europe during the COVID-19 pandemic and its implications on health 

Dear Dr. Bergmann:

I'm pleased to inform you that your manuscript has been deemed suitable for publication in PLOS ONE. Congratulations! Your manuscript is now with our production department. 

Kind regards, 

on behalf of

Professor Mihajlo Jakovljevic 

Academic Editor

PLOS ONE